# Residual Strength Evaluation of Corroded Textile-Reinforced Concrete by the Deep Learning-Based Method

**DOI:** 10.3390/ma13143226

**Published:** 2020-07-20

**Authors:** Wei Wang, Peng Shi, Lu Deng, Honghu Chu, Xuan Kong

**Affiliations:** 1Key Laboratory for Damage Diagnosis of Engineering Structures of Hunan Province, Hunan University, Changsha 410082, China; wang_wei@hnu.edu.cn (W.W.); kongxuan@hnu.edu.cn (X.K.); 2College of Civil Engineering, Hunan University, Changsha 410082, Hunan, China; shipeng@hnu.edu.cn (P.S.); chuhonghu@hnu.edu.cn (H.C.)

**Keywords:** textile-reinforced concrete, deep learning method, faster R-CNN, residual strength evaluation, corrosion degree, microstructure features

## Abstract

Residual strength of corroded textile-reinforced concrete (TRC) is evaluated using the deep learning-based method, whose feasibility is demonstrated by experiment. Compared to the traditional method, the proposed method does not need to know the climatic conditions in which the TRC exists. Firstly, the information about the faster region-based convolutional neural networks (Faster R-CNN) is described briefly, and then procedures to prepare datasets are introduced. Twenty TRC specimens were fabricated and divided into five groups that were treated to five different corrosion degrees corresponding to five different residual strengths. Five groups of images of microstructure features of these TRC specimens with five different residual strengths were obtained with portable digital microscopes in various circumstances. With the obtained images, datasets required to train, validate, and test the Faster R-CNN were prepared. To enhance the precision of residual strength evaluation, parameter analysis was conducted for the adopted model. Under the best combination of considered parameters, the mean average precision for the residual strength evaluation of the five groups of the TRC is 98.98%. The feasibility of the trained model was finally verified with new images and the procedures to apply the presented method were summarized. The paper provides new insight into evaluating the residual strength of structural materials, which would be helpful for safety evaluation of engineering structures.

## 1. Introduction

Textile-reinforced concrete (TRC), which is a new type of composite cement-based material, has received great attention due to its high tensile strength and excellent performance in alkali resistance. Many studies have been carried out for investigating the basic mechanical properties of the TRC. Some scholars investigated the effects of such parameters as the loading rate, temperature, and the arrangement of textile layers on the bending behavior of members made of the TRC through three-point or four-point bending experiments [1,2,3,4]. Some investigated the effects of some parameters, including the prestress levels, steel fiber properties, and freezing-thawing cycles, on the tensile performance of members made of the TRC [5,6]. Kong et al. [7] compared the tensile and flexural behavior of the TRC and found that the ultimate tensile strength of TRC obtained with bending experiments is higher than that obtained with tensile experiments. Additionally, some numerical models were developed for predicting the bending and tensile behaviors of TRC sandwich beams and verified with experiments [8,9,10]. All these studies put emphasis on determining the ultimate tensile strength of newly fabricated TRC components. In fact, accurately evaluating the residual strength of the TRC components is of great importance for their safety assessment. To the best of the authors’ knowledge, there is little literature emphasizing the evaluation of the residual strength of the TRC. Orlowsky and Raupach [11] reviewed a few previous studies on strength loss models of the TRC and then proposed a new numerical model to predict the residual strength of the TRC. However, using these models to predict the residual strength of the TRC one needs to know all the climatic information, including temperature and humidity, which is difficult to obtain in practice.

In recent decades, deep learning models have been developed rapidly and can learn unique features of objects from a great deal of labelled data [12]. Some scholars summarized how some common deep learning models work and analyzed their advantages and disadvantages [13,14]. The deep learning method has also been used in plenty of aspects of civil engineering. The most common application focuses on crack detection, including identifying crack class, location, width, and length [15,16]. Using deep learning methods to detect common damage types of civil structures is also a hot topic [17,18]. Moreover, some researchers proposed deep convolutional neural network (CNN)-based methods to evaluate the fatigue performance of steel members [19], while some developed a deep learning-based framework to uncover those unknown relationships between structural damage and structural responses [20]. Furthermore, a few researchers conducted a design optimization for a truss structure using the deep learning-based method and found its superiority to traditional neural networks [21]. Additionally, a machine vision-based intelligence system was developed by Dick et al. to identify and predict the threats that may result in structural failures [22]. Recently, Hadi and Rigoberto [23] and Ge et al. [24] reviewed the application of deep learning method in structural engineering and material science, respectively. Based on deep learning analysis, Li et al. [25] predicted the modulus of heterogeneous materials and Bastidas-Rodriguez et al. [26] addressed the fracture classification problem of metallic materials. Though the deep learning-based approaches have been applied in many aspects of material science and engineering, up to now no such approaches have been reported to be used to evaluate the residual strength of the TRC in the literature, and the present study offers a first attempt to evaluate the residual strength of materials from a deep learning perspective.

The object of the paper is to evaluate residual strength of the TRC based on a deep learning framework. The paper is divided into six parts. In Section 2, details of the adopted deep learning-based approach, namely, the faster region-based CNN (Faster R-CNN) is introduced. In Section 3, procedures to prepare datasets and to implement the Faster R-CNN are described; indices used to evaluate the performance of trained models are introduced as well. In Section 4, three key parameters that may affect the evaluation accuracy are investigated, based on which an optimal combination of these parameters was obtained to train the Faster R-CNN. The feasibility of the trained model was checked with new images. In Section 5, strategies for enhancing the accuracy of the presented method are discussed and procedures to apply such an approach are summarized. Finally, some remarkable conclusions are made from the study and some future research directions are pointed out.

## 2. Methodology

The Faster R-CNN, which was developed on the basis of the region proposal network (RPN) and fast region-based CNN (Fast R-CNN) both of which utilize the CNN for feature extraction, was adopted to evaluate the residual strength of the TRC in the present study. The Faster R-CNN architecture is illustrated in Figure 1. The RPN plays a role to generate object proposals from input images, while the role of the Fast R-CNN is to determine the location of these object proposals generated by the RPN and evaluate the residual strength.

### 2.1. The CNNs

Generally, a CNN consists of several convolutional (CONV) layers, max pooling (MP) layers, fully-connected (FC) layers, and soft-max (SM) layers. The CONV layers play a role for feature extraction from input images through a group of kernels composed of learnable weights. The depth of kernels is the same as that of the input layer, but the width and height of the former are smaller than that of the latter. Each kernel is set to slide on the input with a specified stride length and at each location of the kernel, as demonstrated in Figure 2, the dot product is carried out between the kernel and its respective field on the input. The stride length has a significant effect on the computation efficiency and the output size. Smaller stride length results in lower computation efficiency and larger output size, and helps to reduce feature loss. The values of the dot product, namely, the element-by-element multiplication between each kernel element and the corresponding element in the respective field, are added up plus a bias as the outcome of each kernel. All the outcomes of each kernel sliding to different locations of the input are arranged as the output of CONV layers. The output size of the CONV layer is affected by the input size, the kernel size as well as the stride length, and may be smaller than the input size. As feature loss may occur due to the size reduction of the output layer, zero padding the input, as shown in Figure 2, is an efficient way to keep the output size.

An MP layer plays a role for size reduction of its input through the operation of downsampling which can save the computation time and reduce the probability of overfitting. Additionally, the MP layer performs an operation of extracting the maximum value from the window that slides on the input, as demonstrated in Figure 3.

An FC layer plays a role for the connection of all neurons from its previous layer, whose role is different from that of a CONV layer that connects neurons of a local region. In fact, the FC layer is a vector consisting of neurons obtained through the operation of the dot product with a bias for each neuron of its inputs.

A SM layer plays a role for the prediction of the category of its input according to the probabilities of the input being each category. The probabilities are computed with an SM function using the features provided by the FC layer. The input is categorized to be the category with the highest probability.

### 2.2. The Region Proposal Network

The RPN, whose overall architecture is demonstrated in Figure 4, plays a role to efficiently generate high-quality region proposals, through sharing CONV layers with the object detection network of the Fast R-CNN adopted in the present study. As can be seen in Figure 4, when an image is fed into the RPN, the output are a number of region proposals that are generated through sliding a mini-size network on the feature map (of the input image) obtained from the last CNN layer. The mini-size network, in fact, is a 3 × 3 spatial window of the feature map in the present study, as suggested by Ren et al. [27]. At the location of each sliding-window nine region proposals with different sizes are produced, which are actually nine rectangular boxes called anchors. These anchors are located at the center of the sliding-window and can be determined with eight parameters (i.e., the sliding-window center (*x_a_*, *y_a_*), three widths and three heights: ($w_{a}^{m}$, $h_{a}^{n}$), where *m*, *n* = 1–3). A concept of Intersection-over-Union (IoU) was proposed to estimate the matching degree between an anchor and a ground-true box (GTB). The IoU of an anchor and a ground-true box (GTB) is calculated to be the ratio of the area of their intersection to the area of their union. An anchor is labelled as positive if it achieves the highest IoU with a GTB, or if its IoU with every GTB is larger than 0.7. A non-positive anchor is labelled as negative if its IoU with any GTB is smaller than 0.3. Anchors that are not labelled are abandoned in the process of training.

For each sliding window, a feature vector is obtained based on the activation function, such as the rectified linear unit (ReLU) function, which is commonly adopted as it not only provides nonlinearity but also enhances the convergence rate. The obtained feature vector is then taken as the input of two correlated functional layers. One functional layer is the box-classification layer, which computes the probability of being an object or just being part of the background in each anchor according to the feature vector and initial weights. The computed probability updates and varies between zero and one during the training process, and eventually gets close to one for a positive anchor and zero for a negative anchor. The other functional layer is the box-regression layer, which computes and updates the parameters that determine the location and size of the predicted bounding box (PBB) associated with an anchor during the process of training to better match a GTB [28].

Training the RPN end-to-end is, as a matter of fact, a process to minimize the loss function shown in Equation (1) using 128 positive and 128 negative anchors selected randomly from an image. The techniques of backpropagation and stochastic gradient descent (SGD) min-batch are employed in the training process:(1)$$L\left(\left\{p_{i}\right\},\left\{t_{i}\right\}\right)=\frac{1}{N_{cls}}\underset{i}{\sum }L_{cls}\left(p_{i},p_{i}^{*}\right)+\lambda \frac{1}{N_{reg}}\underset{j\in \left(x,y,w,h\right)}{\sum }p_{i}^{*}L_{reg}\left(t_{i,j},t_{i,j}^{*}\right)$$

In Equation (1), *L_cls_* and *L_reg_* are the classification loss function and the regression loss function, respectively; *i* represents the numerical order of an anchor in the min-batch, and $p_{i}^{*}$ represents the ground-truth label, which adopts a value of 1 or 0 for positive or negative anchors, respectively; *p_i_* represents the predicted probability of an object in the *i*th anchor. To normalize the two loss functions, the values of *N_cls_* and *N_reg_*, as suggested by Ren et al. [27], were adopted to be the mini-batch size (MBS) and ten percent of the number of anchors, respectively. The *t_i,j_* (*j* = *x, y, w, h*) is a vector that describes geometrical differences between the PBB and the anchor, and the $t_{i,j}^{*}$ is a vector that describes geometrical differences between the GTB and the anchor. The *t_i,j_* and $t_{i,j}^{*}$ can be obtained with the following matrix:(2)$$\left[t_{i,j}\;t_{i,j}^{*}\right]=\left[\begin{array}{c} t_{i,x}\;t_{i,x}^{*}\\ t_{i,y}\;t_{i,y}^{*}\\ t_{i,w}\;t_{i,w}^{*}\\ t_{i,h}\;t_{i,h}^{*} \end{array}\right]=\left[\begin{array}{c} \left(x_{i}-x_{i,a}\right)/w_{i,a}\;\left(x^{*}-x_{i,a}\right)/w_{i,a}\\ \left(y_{i}-y_{i,a}\right)/h_{i,a}\;\left(y^{*}-y_{i,a}\right)/h_{i,a}\\ \log\left(w_{i}/w_{i,a}\right)\;\log\left(w^{*}/w_{i,a}\right)\\ \log\left(h*/h_{i,a}\right)\;\log\left(h^{*}/h_{i,a}\right) \end{array}\right]$$
where (*x_i_*, *y_i_*, *w_i_*, *h_i_*) determines the center location and sizes (width and height) of the PBB associated with the *i*^th^ anchor; Similarly, the team (*x_i,_*_a_, *y_i,a_*, *w_i,_*_a_, h*_i,a_*) determines the center location and sizes of the *i*^th^ anchor, and (*x^*^*, *y^*^, w^*^*, *h^*^*) determines the center location and sizes of the GTB. It should be noted that the four parameters determining the PBB are continuously renewed to approach those of the GTB in the process of training. The position relations among the PBB, the anchor, and the GTB are demonstrated in Figure 5.

In Equation (1), the log loss function, *L_cls_* = −log*p_u_*, was selected as the classification loss function and the Equation (3) shown below was used as the regression loss function:(3)$$L_{reg}\left(y_{1},y_{2}\right)=\{\begin{array}{c} 0.5{(}^{y_{1}}\;if\left|y_{1}-y_{2}\right|<1\\ \left|y_{1}-y_{2}\right|-0.5\;otherwise \end{array}$$
where *y_1_* and *y*_2_ are variables for illustration. For more detailed information on the training process of the RPN can be found in [27].

### 2.3. Fast R-CNN

The Fast R-CNN plays a role for localizing and classifying objects in images and its overall architecture is demonstrated in Figure 6. As shown in Figure 6, the Fast R-CNN also makes use of the CNNs to acquire the feature map of the input image and adopts the object proposals provided by the RPN. Features on the feature map, associated with an object proposal, are usually called a region of interest (RoI). For each RoI, a fixed-size feature vector is acquired through the max pooling operation conducted in the RoI pooling layer (Figure 6). The acquired feature vector is then fed into several FC layers followed by two functional layers. One functional layer is the softmax layer that computes and displays the probability of a RoI being each of g + 1 classes (g training categories +1 background category), and the other functional layer is the regression layer that computes and displays the four parameters that determine the center location ($T_{x}^{u}$,$T_{y}^{u}$), height ($T_{h}^{u}$), and width ($T_{w}^{u}$) of object bounding boxes. The IoU of the RoI and the GTB is also used to estimate their matching degree. For each RoI, it is labelled as positive (*u* = 1) when its IoU with a GTB is greater than 0.5, and it is labelled as negative (*u* = 0) when the maximum value of its IoU with all the GTB is in the range of [0.1,0.5) [27].

Training the Fast R-CNN end-to-end is, in fact, a process to minimize the multi-class loss function given in Equation (4) for each labelled RoI, in which the techniques of backpropagation and the SGD min-batch are employed:(4)$$L\left(p,u,T^{u},v\right)=L_{cls}\left(p,u\right)+\lambda \left[u\geq 1\right]\underset{i\in \left\{x,y,w,h\right\}}{\sum }L_{reg}(T_{i}^{u},v_{i})$$

In Equation (4), *L_cls_* = −log*p_u_* stands for the log loss function, and *L_reg_* stands for the regression loss function as given in Equation (3); *u* is the label of the GTB, and *v* is a vector that determines the location coordinates and sizes (height and width) of the GTB. The Iverson bracket [u ≥ 1] adopts a value of 1 when u ≥ 1 and 0 otherwise. To keep balance between the two loss functions, the hyper-parameter λ adopted a value of 1 [27]. During each iteration, two images and 128 RoIs (consisting of mini-batches) acquired from the two images are picked at random to train the Fast R-CNN. From the study of Girshick et al. [28], readers can find more detailed information on the training process of the Fast R-CNN.

### 2.4. Architecture of the CNNs Based on VGG16-Net

To enhance computing efficiency, the RPN and Fast R-CNN are intentionally designed to use the same architecture as the CNN. Currently, many famous architectures have been developed for the CNN, including the Microsoft ResNet-152, GoogleNet, ZF-Net, and VGG16-net [18]. As the VGG16-net can make a good balance between the computing efficiency and detecting accuracy, therefore, it was chosen for the CNN architecture in the present study. The VGG16-net is usually constituted by thirteen weighted CONV layers, five MP layers, three weighted FC layers, and a SM layer. All CONV layers make use of nonlinear activation functions (i.e., the ReLU) to enhance the convergence rate, and take advantage of the technique of zero-padding to maintain their spatial sizes. All MP layers, also using zero-padding to maintain size, conduct a spatial pooling operation by sliding 2 × 2 filters two pixels per stride. Following the CONV and MP layers are three FC layers and a SM layer which is adopted to classify objects in images.

To present a Faster R-CNN-based framework to evaluate the residual strength of the TRC, modification was made to the initial overall architecture of VGG16-net to better match the RPN and Fast R-CNN. With regard to the modified RPN demonstrated in Figure 7, the final MP layer and the three FC layers of the primary VGG16-net was substituted with a sliding CONV after which there is an FC layer with 512 dimensions in depth, and the SM layer of the primary VGG16-net was substituted with the SM and regression layers. The detailed information about the VGG16-net-based RPN is summarized in Table 1.

With regard to the modified Fast R-CNN demonstrated in Figure 8, the final MP layer of the primary VGG16-net was substituted with a RoI pooling layer. For the purpose of preventing overfitting during the process of training, between each of the three FC layers of the primary VGG16-net were inserted with dropout layers whose threshold value was set to be 0.5. To match the number of classifications considered in the present study, the depth of the final FC layer was, thus, altered correspondingly to six for five residual strengths and background. The final SM layer was substituted with the SM and regression layers. Table 2 summarizes the detailed information about the VGG16-net-based Fast R-CNN.

### 2.5. Faster R-CNN Composed of the RPN and Fast R-CNN

To improve computing speed efficiently, the Faster R-CNN were designed intentionally to combine the RPN and Fast R-CNN that share the same CNNs for image feature extraction, as demonstrated in Figure 9. Training the Faster R-CNN is actually a four-step alternating process. Step 1 is to train the RPN following the procedures discussed in Section 2.2, in which the object proposals to be used for training the Fast R-CNN are prepared. The second step is to train the Fast R-CNN, following these procedures discussed in Section 2.3, with the object proposals prepared in step 2. The third step is to initialize the RPN with the final weights obtained from previous step, and to fine-tune these layers exclusive to the RPN with the shared CONV layers fixed. The final step is to fine-tune these layers exclusive to the Fast R-CNN utilizing the object proposals obtained in step 3 with the shared CONV layers fixed. As hundreds to thousands of object proposals are generated from an image through the RPN, which will lower the computing efficiency and estimating accuracy, these object proposals are sorted based on the scores obtained from the box-classification layer, and the first 2000 object proposals are utilized for the training of the Fast R-CNN in step two. Additionally, it has been proved that training the Faster R-CNN with the first 300 object proposals obtained from the final step can make a good balance between detecting accuracy and detecting speed.

## 3. Dataset Preparation and Implementation Details

### 3.1. Dataset Preparation

To estimate the residual strength of the TRC based on deep learning approaches, datasets need to be prepared beforehand to train, validate, and test the Faster R-CNN model. As demonstrated in Figure 10, datasets were generated following a three-step procedure in the present study.

In the first step, specimens of the TRC were prepared. Twenty TRC specimens were fabricated and cured under standard conditions in Shunxing Concrete Co. LTD in Hunan Province of China. These TRC specimens were evenly divided into five groups. One group was exposed to normal conditions while other four groups were treated to four different corrosion degrees by immersing them into the tank made by Cangzhou Xingye Test Instrument Co. LTD in Hebei Province of China, which contained water at 80 degrees, for different days, as demonstrated on the left of Figure 10. The immersed time for the five groups of specimens was 0, 6, 12, 18, and 24 days, respectively, and the corresponding corrosion degrees were 0, 0.43, 0.48, 0.52, and 0.56, respectively, which corresponded to five different residual strengths of 1, 0.57, 0.52, 0.48, and 0.44, respectively, that were denoted with P-0, P-6, P-12, P-18, and P-24, respectively [29]. It should be noted that in the present study residual strengths were obtained from three-point bending test and the value of residual strength under each corrosion degree is the average residual strength of the four specimens under consideration.

In the second step, images of the TRC specimens were captured under different corrosion degrees. At each corrosion degree, two-megapixel portable digital microscopes with a brand of Smolia made in Fukuoka, Japan (1920 × 1080) were used to capture the microstructure features of the TRC specimens, as demonstrated in the middle of Figure 10. As the field of view of the portable digital microscope is only 2.3 × 1.3 mm, the resolution of the captured images reaches approximately 21,000 dots per inch (dpi). To enhance the robustness of the estimation, different portable digital microscopes with the same specification were adopted, based on which images were captured under different light conditions by different photographers. Under each corrosion degree, 550 initial images were taken from all specimens, with 500 images used to train the model and 50 reserved for robustness verification of the trained model. After augmenting the dataset via the operation of mirror (including horizontal and vertical) and 180-degree rotation, the number of images used for model training and robustness verification increased to 10,000 (that is, 500 × 5 × 4) and 1000 images (that is, 50 × 5 × 4), respectively.

In the final step, datasets were established utilizing images obtained in step 2. As the Faster R-CNN is a supervised model, images need to be labelled first and then used for training the model. The 10,000 images (2000 for each residual strength) obtained for model training in step two were labelled with the days that they were immersed in the water, as demonstrated on the right of Figure 10. For instance, if a specimen was immersed into the water for six days, its corresponding image was labeled to be P-6. Among the 10,000 labelled images, the proportion of images utilized for creating training, validating, and testing datasets were 40%, 40%, and 20%, respectively.

### 3.2. Implementation Details

Based on the open-source Faster R-CNN model, all experiments were implemented on a service station with the Caffe framework under the graphics processing unit (GPU) mode. The hardware configuration of the service station is as follows: central processing unit (CPU): Intel i7-8700k (3.20 GHz), GPU: 11 GB memory, 16 GB DDR4 memory, and a ZOTAC X-GMING GeForce RTX 2080Ti. For Faster R-CNN, the sizes of all images adopted for training and validating the RPN and Fast R-CNN are resized to make the maximum value of their long and short sides smaller than 1000 and 600 pixels, respectively. The original parameters for the CNN layers and FC layers are obtained from two zero-mean Gaussian distributions whose standard deviations are 0.001 and 0.01, respectively. The values of the MBS, learning rate (LR), momentum, and weight decay adopted to train the RPN and Fast R-CNN are 128, 0.001, 0.9, and 0.0005, respectively. The scales of nine anchors are obtained by combining three different scales {128^2^, 256^2^, 512^2^} and three different aspect ratios {1:1, 1:2, 2:1}. Since cross-boundary anchors could inevitably lead to a non-convergence problem, anchors whose boundaries cross images were abandoned in the process of training. Additionally, the value of non-maximum suppression was given a value of 0.7 for the reduction of overlap between object proposals. A more detailed description about parameter initialization of the Faster R-CNN can be found in [27].

Average precision (AP), which is calculated on the basis of the precision-recall curve of each class, is used for estimating the performance of the trained Faster R-CNN model [30]. For each class, the definition of precision is the proportion of correct detections to all the detections returned by the algorithm, and the definition of recall is the proportion of correct detections to all the considered ground-truth instances. The terminology mean AP (mAP), as the term suggests, is the mean of all calculated APs. From the study of Girshick [30], readers can find more details about the precision-recall curve and the AP.

## 4. Experiments

### 4.1. Training, Validating, and Testing Results

The Faster R-CNN model was first trained through the previously discussed four-step training strategy using original parameters and was tested using the testing dataset. With the service station introduced in Section 3.2, the time required for training the model for 280,000 iterations and evaluating an image with a resolution of 1920 × 1080-pixel is around 14.0 h and 0.072 s, respectively. Figure 11 shows the change of training loss against the number of iterations. As can be observed from Figure 11, the training loss declines as the number of iterations increases and gradually becomes stable after 230,000 iterations. Figure 12 shows the precision-recall curve of each case under consideration for the testing dataset based on the model trained for 230,000 iterations. With the obtained precision-recall curve, the APs and mAP can be computed, as also shown in Figure 12. It can be observed from Figure 12 that the APs for residual strength evaluation of P-0, P-6, P-12, P-18, and P-24 are 99.51%, 99.75%, 98.50%, 90.43%, and 90.75%, respectively, and the relevant mAP is 95.79%. Likewise, based on the models trained for different iterations, the APs and mAPs were obtained and plotted against the number of iterations, as shown in Figure 13. Expectedly, as the number of iterations increases, the APs and mAPs both increase at first and then tend to be stable after 230,000 iterations.

### 4.2. Parameter Optimization

As the original parameters adopted for Faster R-CNN in the study of Ren et al. [27] might not be the best combination for the dataset under consideration, in the present study parameters for Faster R-CNN were optimized for achieving better accuracy of residual strength evaluation. As indicated by Ren et al. [27] that the performance of Faster R-CNN is significantly affected by three key parameters, namely, the anchor scale (AS), the MBS, and LR. Therefore, the influences of these parameters on the accuracy of residual strength evaluation were investigated. With three sets of ASs, five MBSs, and three LRs considered, the number of combinations of the three parameters was 45. The APs and mAPs calculated from the testing dataset under the 45 combinations were summarized, as illustrated in Figure 14 and Table 3. It should be noted that the three sizes of anchors under consideration were determined based on the size of the images (600 × 1000 pixels) to be detected. It should be pointed out that in the study of Ren et al. an RoI was labelled as the computed category if a probability of less than 0.6 was computed from the SM layer for the RoI [27]. It should also be noted that the number of iterations adopted was to be 230,000 for each model training.

As can be observed from Figure 14 and Table 3, the AS, MBS, and LR affects the APs and mAPs in a coupled way. The largest APs for residual strength evaluation of P-0, P-6, P-12, P-18, and P-24 are 99.93% for Case 34, 99.30% for Case 1, 99.45% for Case 34, 97.92% for Case 23, and 99.08% for Case 23, respectively. To make a better balance among APs for different cases, Case 23 was selected, in which the mAP reaches the largest value of 98.98%, and the corresponding APs for residual strength evaluation of P-0, P-6, P-12, P-18, and P-24 are 99.20%, 99.66%, 99.05%, 97.92%, and 99.08%, respectively. The ASs in Case 23 are 128^2^, 256^2^, and 512^2^, and the anchor ratios are 1:1, 1:2, and 2:1. The MBS and LR in Case 23 are 64 and 0.0005, respectively.

### 4.3. Testing New Images

To examine the feasibility of the presented method, the model obtained in Case 23 was used for evaluating the residual strength of the 1000 images (200 for each residual strength) that were reserved in Section 3.1. The evaluation results for the five cases under consideration are summarized in Table 4, from which it can be seen that the APs for residual strength evaluation of P-0, P-6, P-12, P-18, and P-24 are 99.5%, 99.5%, 100%, 100%, and 100%, respectively, and the corresponding mAP is 99.8%. The results demonstrate that the trained model also has an excellent performance for residual strength evaluation of new images, demonstrating the feasibility of the presented method. Figure 15 illustrates some of the evaluation results for each case under consideration, in which the images that were wrongly evaluated for residual strength of P-0 and P-6 were given specifically in Figure 15a,b. It should be pointed out that the 1000 reserved new images were captured in various circumstances as discussed in Section 3.1, which means that the effects of these circumstances on the accurate rate of residual strength evaluation is insignificant. It should also be pointed out that during the testing process for each new image, the well-trained model will output a predicted value of residual strength represented by the image. if the predicted value matches the actual residual strength that was obtain from experiment, the evaluation is assumed to be correct and vice versa. The average precision presented in Table 4 was calculated to be the ratio of the number of correct evaluations to the total evaluations.

## 5. Discussion

The results illustrated in Section 4 have demonstrated that the proposed Faster R-CNN-based approach is capable of learning and detecting microstructure feature differences of the TRC with different residual strengths. In reality, the actual residual strength required to be estimated is usually different from one of those adopted for training the model. Therefore, the actual residual strength is most likely to be evaluated to be the training residual strength which is closest to the actual residual strength. To reduce the evaluation error, one effective approach is to use more specimens for training the model, which contributes to shortening the gap between the actual residual strength and the closest training residual strength. Additionally, the surface of the TRC component required for residual strength evaluation needs to be cleaned to take images for good quality. It should be pointed out that different types of materials own different microstructure features, therefore, new models are required to train for different types of materials. The procedures for implementing the proposed deep learning-based framework for residual strength evaluation of a material are summarized below:Prepare sufficient samples under differing corrosion degrees and guarantee that the samples could contain the scope of residual strengths required to be evaluated and thus enhance the evaluation precision.Acquire high-quality images of prepared samples and label acquired images.Select a proper deep learning-based framework and train it using these images acquired in step 2; check the soundness of the trained model with new images not adopted for training.Acquire the images of components that need to be evaluated and utilize the trained model to estimate the residual strength.

It should be pointed out that enhancing image quality and augmenting the dataset of images adopted for training and validating the model are efficient ways to improve the model performance.

## 6. Conclusions

Previous models to predict the residual strength of textile-reinforced concrete need to know the climatic conditions (temperature and humidity) in which the TRC exists, which is difficult in practice. A deep learning-based framework based on the Faster R-CNN is presented to evaluate the residual strength of the TRC under different corrosion degrees, without the need to know the climatic conditions. Five groups of TRC specimens were fabricated and treated to five different corrosion degrees corresponding to five different residual strengths by immersing them into the water tank for different days (namely, from 0 to 24 days with an interval of six days, denoted with P-0, P-6, P-12, P-18, and P-24, respectively). Images of microstructure features of these specimens with five different residual strengths were taken in various circumstances with portable digital microscopes. The resolution of the obtained images reaches approximately 21,000 dots per inch (dpi). Among the 11,000 images adopted in the study, the proportion of 10,000 images utilized for creating the training, validating, and testing datasets were 40%, 40%, and 20%, respectively, and the other 1000 new images (200 for each strength) were reserved to check the feasibility of the trained models.

The influences of three key parameters, namely, the anchor scale, mini-batch size, and learning rate, on the precision of residual strength evaluation were investigated, based on which a best combination of the three parameters was acquired to train the Faster R-CNN. The maximum APs for residual strength evaluation of P-0, P-6, P-12, P-18, and P-24, obtained under the best combination of these parameters, are 99.20%, 99.66%, 99.05%, 97.92%, and 99.08%, respectively, and the mAP is 98.98%. It should be noted that the maximum APs were obtained by make comparisons with the results obtained from experiments.

The paper provides a new way to evaluate residual strength of materials. However, it should be pointed out that under the same corrosion degree the microstructure features of different materials differ from each other. This indicates that specific models are required for different materials. In the future, efforts will be focused on other types of materials which are widely utilized in industry to further check the feasibility of the presented approach. It should also be noted that the presented method is actually a way to build the relationship between the microstructure features and micro properties of a material, which can be applied in many fields.

## Figures and Tables

**Figure 1 materials-13-03226-f001:**
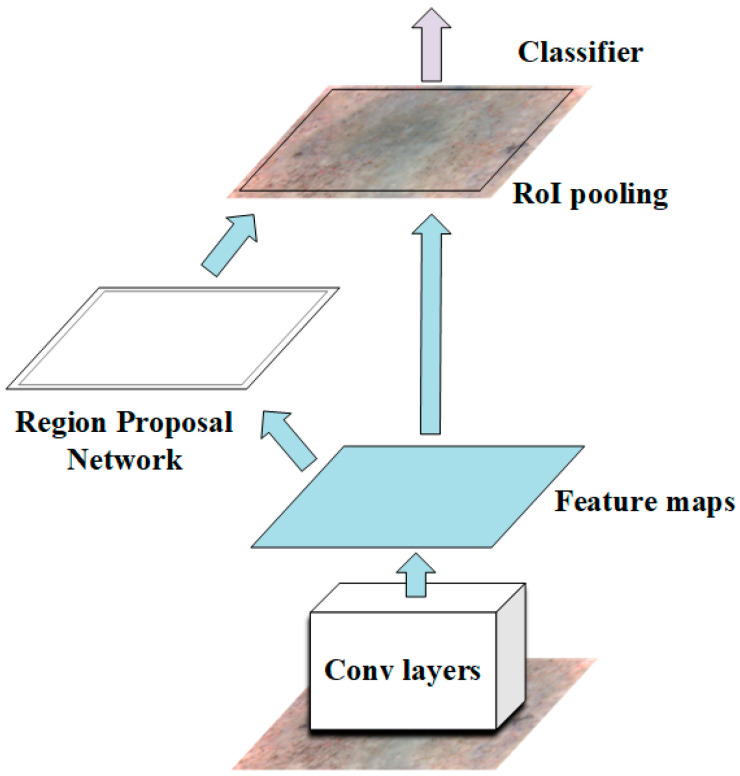
Architecture of the Faster R-CNN.

**Figure 2 materials-13-03226-f002:**
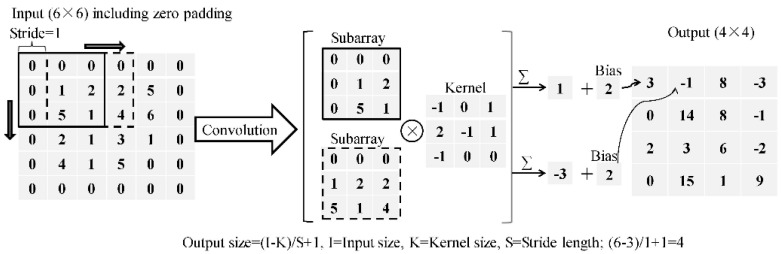
Demonstration of the convolutional layer.

**Figure 3 materials-13-03226-f003:**
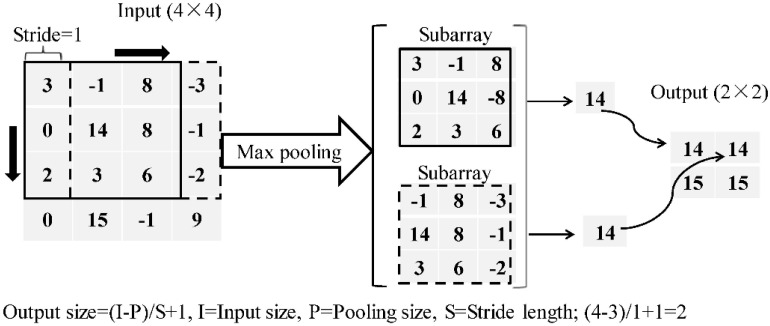
Demonstration of the max-pooling layer.

**Figure 4 materials-13-03226-f004:**
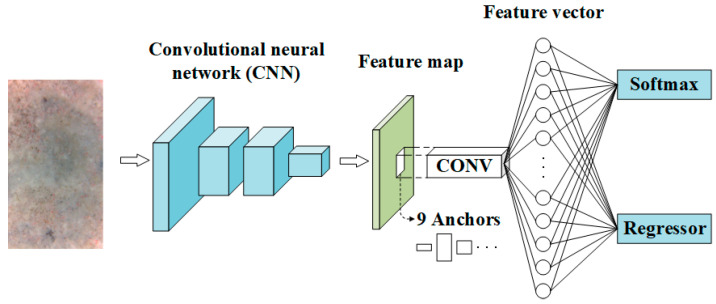
Overall architecture of the original RPN.

**Figure 5 materials-13-03226-f005:**
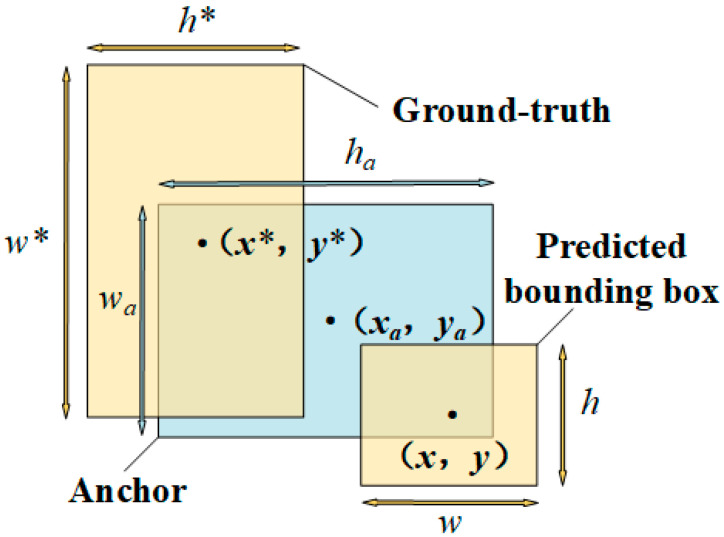
Illustration of position relations among the PBB, the anchor and the GTB.

**Figure 6 materials-13-03226-f006:**
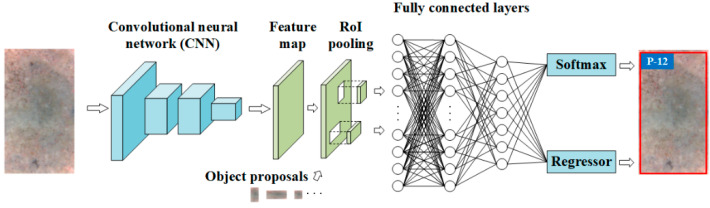
Overall architecture of the original Fast R-CNN.

**Figure 7 materials-13-03226-f007:**
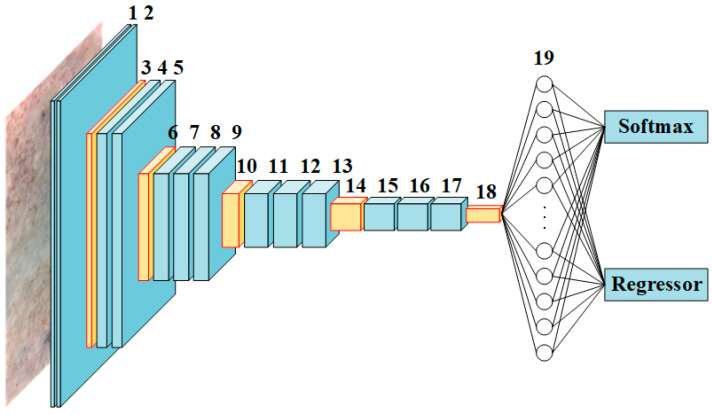
Overall architecture of the modified RPN based on VGG-16.

**Figure 8 materials-13-03226-f008:**
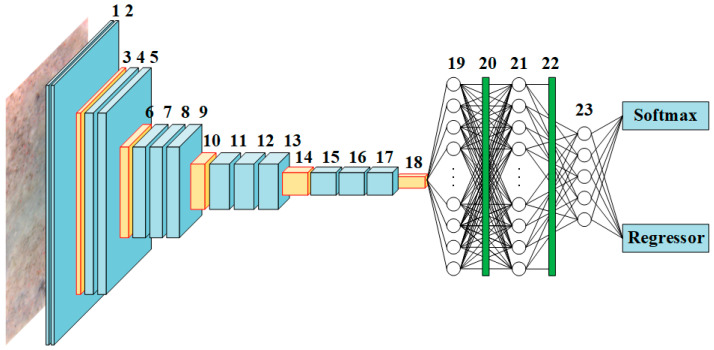
Overall architecture of the modified Fast R-CNN based on VGG-16.

**Figure 9 materials-13-03226-f009:**
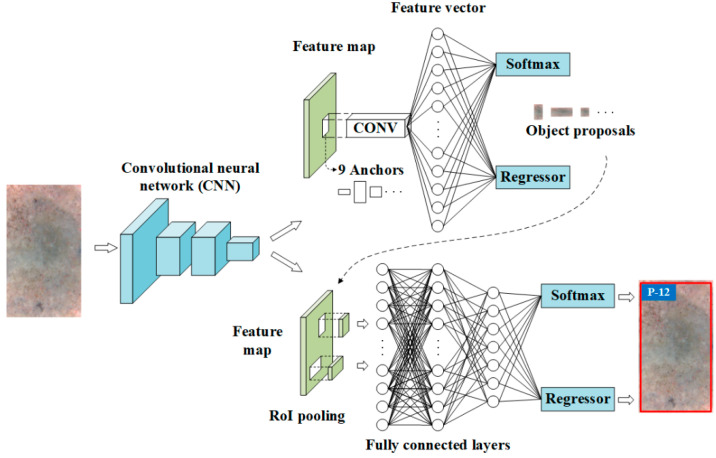
Overall architecture of the Faster R-CNN.

**Figure 10 materials-13-03226-f010:**
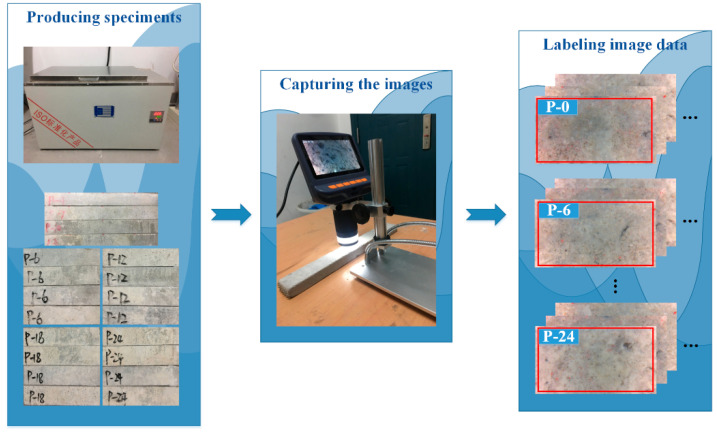
Procedures to generate datasets.

**Figure 11 materials-13-03226-f011:**
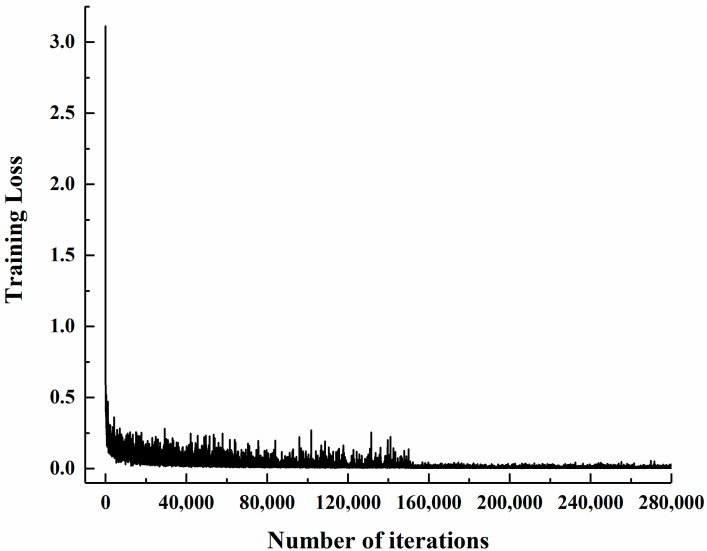
Change of training loss against the number of iterations.

**Figure 12 materials-13-03226-f012:**
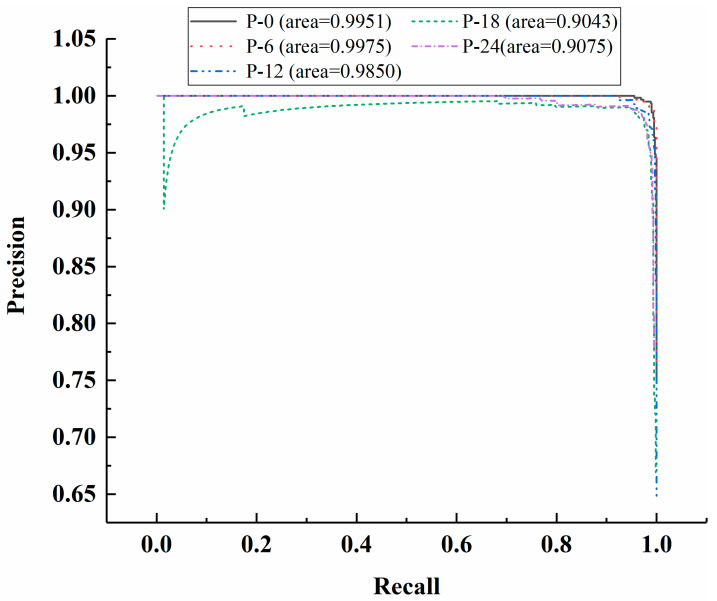
Precision-recall curve of each case under consideration for the testing dataset based on the model trained for 230,000 iterations.

**Figure 13 materials-13-03226-f013:**
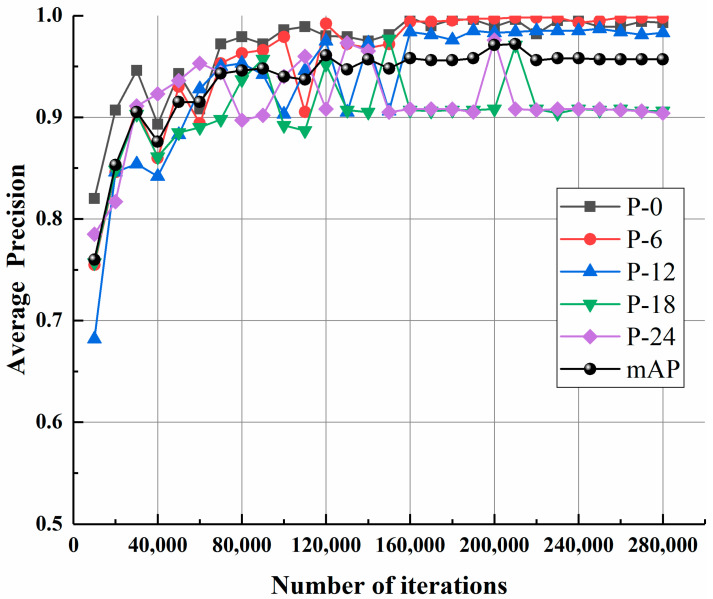
Variation of APs and mAPs against the number of iterations.

**Figure 14 materials-13-03226-f014:**
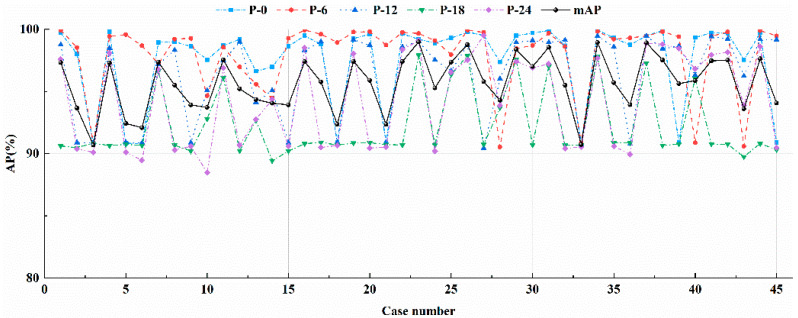
APs and mAPs under different combinations of the AS, MBS, and LR.

**Figure 15 materials-13-03226-f015:**
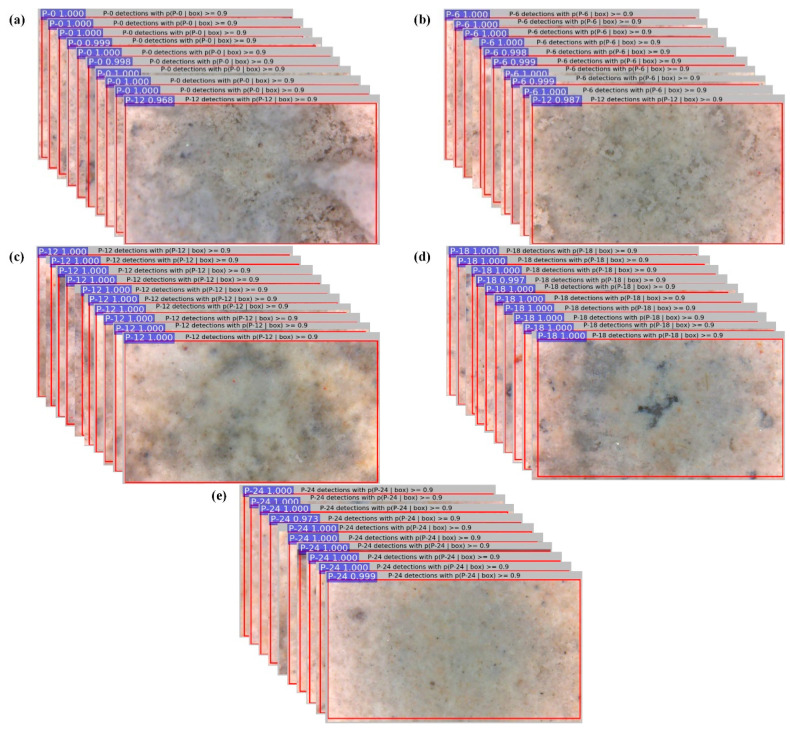
Illustration of some of the evaluation results for each case under consideration: (**a**) P-0; (**b**) P-6; (**c**) P-12; (**d**) P-18; (**e**) P-24.

**Table 1 materials-13-03226-t001:** Detailed information about the VGG16-net-based RPN.

Layer	Type	Filter Size	Stride	Depth	Layer	Type	Filter Size	Stride	Depth
1	CONV + ReLU	3 × 3	1	64	11	CONV + ReLU	3 × 3	1	512
2	CONV + ReLU	3 × 3	1	64	12	CONV + ReLU	3 × 3	1	512
3	Max pooling	2 × 2	2	64	13	CONV + ReLU	3 × 3	1	512
4	CONV + ReLU	3 × 3	1	128	14	Max pooling	2 × 2	2	512
5	CONV + ReLU	3 × 3	1	128	15	CONV + ReLU	3 × 3	1	512
6	Max pooling	2 × 2	2	128	16	CONV + ReLU	3 × 3	1	512
7	CONV + ReLU	3 × 3	1	256	17	CONV + ReLU	3 × 3	1	512
8	CONV + ReLU	3 × 3	1	256	18	Sliding CONV + ReLU	-	-	512
9	CONV + ReLU	3 × 3	1	256	19	FC	-	-	512
10	Max pooling	2 × 2	2	256	20	Softmax and Regressor	-	-	-

**Table 2 materials-13-03226-t002:** Detailed information about the VGG16-net-based Fast R-CNN.

Layer	Type	Filter Size	Stride	Depth	Layer	Type	Filter Size	Stride	Depth
1	CONV + ReLU	3 × 3	1	64	13	CONV + ReLU	3 × 3	1	512
2	CONV + ReLU	3 × 3	1	64	14	Max pooling	2 × 2	2	512
3	Max pooling	2 × 2	2	64	15	CONV + ReLU	3 × 3	1	512
4	CONV + ReLU	3 × 3	1	128	16	CONV + ReLU	3 × 3	1	512
5	CONV + ReLU	3 × 3	1	128	17	CONV + ReLU	3 × 3	1	512
6	Max pooling	2 × 2	2	128	18	RoI pooling	-	-	512
7	CONV + ReLU	3 × 3	1	256	19	FC + ReLU	-	-	4096
8	CONV + ReLU	3 × 3	1	256	20	Dropout	-	-	-
9	CONV + ReLU	3 × 3	1	256	21	FC + ReLU	-	-	4096
10	Max pooling	2 × 2	2	256	22	Dropout	-	-	-
11	CONV + ReLU	3 × 3	1	512	23	FC + ReLU	-	-	7
12	CONV + ReLU	3 × 3	1	512	24	Softmax and Regressor	-	-	-

**Table 3 materials-13-03226-t003:** Parameters considered and corresponding results for test dataset.

Cases	Anchor Scales Aspect Ratio	Mini-Batch Sizes	Learning Rate	AP (%)					mAP (%)
				P-0	P-6	P-12	P-18	P-24	
1	{64^2^,128^2^,256^2^} {1:1,1:2,2:1}	16	0.0001	99.70	99.93	98.76	90.61	97.59	97.32
2			0.0005	98.02	98.53	90.89	90.48	90.37	93.66
3			0.001	90.70	90.86	90.89	90.84	90.12	90.68
4		32	0.0001	99.81	99.43	98.48	90.64	98.09	97.29
5			0.0005	90.83	99.57	90.91	90.75	90.10	92.43
6			0.001	90.81	98.67	90.86	90.65	89.46	92.09
7		64	0.0001	98.95	97.23	97.20	96.63	96.78	97.36
8			0.0005	98.98	99.21	98.32	90.68	90.29	95.50
9			0.001	98.63	99.27	90.89	90.21	90.58	93.92
10		128	0.0001	97.54	94.66	95.07	92.80	88.48	93.71
11			0.0005	98.68	98.55	97.47	96.10	96.88	97.54
12			0.001	99.20	96.97	98.99	90.23	90.68	95.21
13		256	0.0001	96.62	95.58	94.12	92.66	92.75	94.35
14			0.0005	96.97	94.46	95.09	89.43	94.36	94.06
15			0.001	98.63	99.27	90.89	90.21	90.58	93.92
16	{128^2^,256^2^,512^2^} {1:1,1:2,2:1}	16	0.0001	99.48	99.93	98.31	90.80	98.50	97.40
17			0.0005	98.79	99.60	99.00	90.91	90.49	95.76
18			0.001	90.91	98.93	90.91	90.69	90.65	92.36
19		32	0.0001	99.26	99.77	99.16	90.85	98.04	97.42
20			0.0005	99.60	99.80	98.72	90.88	90.46	95.89
21			0.001	90.88	98.73	90.91	90.73	90.51	92.35
22		64	0.0001	99.63	99.75	98.58	90.69	98.38	97.41
23			0.0005	99.20	99.66	99.05	97.92	99.08	98.98
24			0.001	98.89	99.08	97.53	90.71	90.20	95.28
25		128	0.0001	99.30	97.96	96.66	96.36	96.53	97.36
26			0.0005	99.77	99.93	98.72	97.86	97.53	98.76
27			0.001	99.51	99.75	98.50	90.43	90.75	95.79
28		256	0.0001	97.36	90.53	96.02	93.68	93.86	94.29
29			0.0005	99.48	98.44	98.95	97.57	97.41	98.37
30			0.001	99.70	98.70	99.10	90.70	96.90	97.00
31	{256^2^,512^2^,1024^2^} {1:1,1:2,2:1}	16	0.0001	99.90	99.68	98.95	97.01	97.22	98.55
32			0.0005	98.67	98.61	99.12	90.67	90.41	95.49
33			0.001	90.81	90.86	90.72	90.69	90.55	90.73
34		32	0.0001	99.93	99.82	99.45	97.79	97.66	98.93
35			0.0005	99.30	99.21	98.58	90.88	90.59	95.71
36			0.001	98.75	99.30	90.77	90.88	89.95	93.93
37		64	0.0001	99.47	99.47	99.45	97.28	98.88	98.91
38			0.0005	99.90	99.80	98.43	90.66	98.80	97.52
39			0.001	90.89	99.40	98.64	90.77	98.46	95.63
40		128	0.0001	99.34	90.88	96.32	96.04	96.80	95.87
41			0.0005	99.71	99.44	99.42	90.77	97.95	97.46
42			0.001	99.67	99.78	99.22	90.76	98.16	97.52
43		256	0.0001	97.54	90.59	96.24	89.74	93.88	93.6
44			0.0005	99.71	99.88	99.18	90.81	98.59	97.63
45			0.001	90.91	99.47	99.16	90.32	90.48	94.07

**Table 4 materials-13-03226-t004:** Evaluation results for new images.

Item	Number of New Images to Be Evaluated	Number of Incorrect Evaluations	AP (%)	mAP (%)
P-0	200	1	99.5	99.8
P-6	200	1	99.5	
P-12	200	0	0	
P-18	200	0	0	
P-24	200	0	0	

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
