# Peer review of "Residual Strength Evaluation of Corroded Textile-Reinforced Concrete by the Deep Learning-Based Method"

_materials, 2020, doi:10.3390/ma13143226_

Round 1

Reviewer 1 Report

The article presents the application of deep learingn-based methods for material evaluation. The topic is up-to-date and can be interesting for the readers but I have some general remarks before I recommend the publication of the paper:

My main concern is the originality of the paper. Deep learning is commonly used in material evaluation. Could you indicate the original elements of the paper in the Introduction?

Deep learning is very popular last years. In my opinion some basic papers about application of deep learning in materials evaluation should be mentioned in the Introduction.

In Section "Discussion" authors claim that the deep learning-based algorithm of residual strength evaluation invloves preparing samples, taking photos and training the model. In real cases it is usually impossible to collect photos or data for unadamages structure/element/material. Could you try to explain how your method can be applied in the real cases if the reference data are not accessible?

Section 2.2. The Title of the section 2.2. should not be writen with the use of abbreviation.

In Introduction Authors wrtite the sentence: "Recently, Hadi and
Rigoberto reviewed the applications of deep learning in structural engineering" - please, add the reference.

Reviewer 2 Report

Some aspects need to be clarified:

  1. Which residual strength is involved in the paper? Tensile, flexural etc.?
  2. In order to establish the efficiency of one method, the comparison have to be made between results obtained with at least two different methods (for example deep learning method and mechanical testing in the laboratory). The paper presents only results obtained with deep leaning method. Which are the control results and how the comparison was made?
  3. Residual strength was determined by experiments? Authors can provide results obtained with any other method in order to make a proper comparison?
  4. Various concrete specimens (made with the same recipe) have different strengthts. Average precision presented in table 4 sounds too good to be true for different concrete specimens. Please specify how you decided if an evaluation is correct or incorrect (citerium for right or wrong).
  5. Conclusions have to be rewritten in order to include residual strength obtained with other methods and the comparison.

Also change 1 with 2 on College of Civil Engineering (under the authors).

Reviewer 3 Report

"The paper deals with a deep learning framework based on Faster R-CNN to evaluate the residual strength of the TRC having different corrosion degrees. The topic treated in the paper is very interesting and well presented. Also, the English language is good. Probably, only the conclusion should be widened. In any case, in the reviewer's opinion, the paper can be published in the journal as in the present form."

Round 2

Reviewer 1 Report

Thank you for response to reviewer's remarks.

Reviewer 2 Report

The revised paper can be published as it is.